# Weak evidence for heritable changes in response to selection by aphids in *Arabidopsis* accessions

Marc W Schmid[1†], Klara Kropivšek[2,3†‡], Samuel E Wuest[2,3,4§], Bernhard Schmid[3,4,5], Ueli Grossniklaus[2,3]*

[1]MWSchmid GmbH, Glarus, Switzerland; [2]Department of Plant and Microbial Biology, University of Zurich, Zurich, Switzerland; [3]Zurich-Basel Plant Science Center, University of Zurich, ETH Zurich, University of Basel, Zurich, Switzerland; [4]Department of Evolutionary Biology and Environmental Studies, University of Zurich, Zurich, Switzerland; [5]Department of Geography, University of Zurich, Zurich, Switzerland

*For correspondence:
grossnik@botinst.uzh.ch

†These authors contributed equally to this work

Present address: ‡Laboratory for Environmental and Life Sciences, University of Nova Gorica, Nova Gorica, Slovenia; §Agroscope, Wädenswil, Switzerland

## eLife Assessment

This article examines selection on induced epigenetic variation ('Lamarckian evolution') in response to herbivory in *Arabidopsis thaliana*. The authors find weak evidence for such adaptation, which contrasts with a recently published study that reported extensive heritable variation induced by the environment. The authors **convincingly** demonstrate that the findings of the previous study were confounded by mix-ups of genetically distinct material, so that standing genetic variation was mistaken for acquired (epigenetic) variation. Given the controversy surrounding the influence of heritable epigenetic variation on phenotypic variation and adaptation, this study is an **important**, clarifying contribution; it serves as a timely reminder that sequence-based verification of genetic material should be prioritized when either genetic identity or divergence is of importance to the conclusions.

**Abstract** In plants, transgenerational inheritance of certain epialleles has been observed but experimental evidence for selection of epigenetic variation independent of genetic variation is scarce. We extended an experiment simulating selection in response to herbivory in *Arabidopsis thaliana* to assess a potential contribution of epigenetic variation to the selected phenotypes within three accessions. To minimize maternal effects, we grew offspring from replicate populations and their ancestors for two generations in a common environment and assessed the phenotypes in the second generation. We found weak evidence for the selection of epigenetic variation: bolting time differed significantly in one accession. Significant differences between maternal lines suggested random residual or novel genetic and/or epigenetic variation. Our results are in conflict with those of a recent study reporting that environment-induced heritable variation is common in *Arabidopsis*. Reanalyzing the data from that study showed that the reported findings resulted from a mix-up of accessions and, thus, reflected genetic rather than epigenetic variation between accessions. To avoid future misinterpretations of studies investigating epigenetic inheritance, we provide guidelines to design experiments that clearly differentiate between epigenetic and genetic variation and distinguish standing variation from de novo variation acquired during an experiment.

## Introduction

Modifications of DNA and histones represent epigenetic marks associated with active or repressed gene expression and play an important role in a plant's development and its responsiveness to the environment. Unlike in animals, plant germlines are not set aside early in development but form later, when somatic cells get committed to eventually form gametes (*Schmidt et al., 2015*). As a result, epigenetic marks affected by environmental conditions can potentially be inherited. Therefore, epigenetic marks may contribute to adaptive responses in changing environments. Furthermore, spontaneous changes in DNA methylation, a mark associated with some naturally arising heritable epialleles in the plant model *Arabidopsis thaliana* (e.g., *Soppe et al., 2000*; *He et al., 2018*; *Schmid et al., 2018b*), occur much more frequently than genetic mutations (*van der Graaf et al., 2015*; *Johannes and Schmitz, 2019*; *Hazarika et al., 2022*). This suggests that the rates of epimutations might be sufficiently high, differentiating epigenetic from genetic variation, yet low enough to maintain a heritable response to selection.

The potential role of epigenetic variation in adaptation depends on several factors: the rate of spontaneous epimutations, the influence of environmental effects on epigenetic variation, the heritability of epimutations, and the impact of epigenetic variation on the phenotype. Importantly, to contribute to adaptation, epigenetic variation must be sufficiently stable, that is, heritable over several generations, so it can be subject to selection. In non-model species, DNA methylation patterns assessed with reduced representation sequencing methods were frequently found to be associated with distinct environments or treatments. However, epigenetic variation was generally correlated with genetic variation (*van Moorsel et al., 2019*; *Mounger et al., 2021*; *Boquete et al., 2022*; *Mounger et al., 2022*; *Ibañez et al., 2023*). Likewise, high-resolution genome-wide studies in *Arabidopsis* have revealed extensive variation in DNA methylation patterns among different populations, but this variation was mostly associated with underlying genetic differences (*Dubin et al., 2015*; *Kawakatsu et al., 2016*). Nonetheless, studies on epigenetic recombinant inbred lines (epiRILs) suggest that epialleles can significantly contribute to phenotypic variation in plants (*Cortijo et al., 2014*; *Kooke et al., 2015*). Thus, while there are examples of pure epialleles *Paszkowski and Grossniklaus, 2011* whose frequency is not related to genetic variation, the role of spontaneous epigenetic variation in adaptation remains controversial (*Jablonka and Raz, 2009*; *Quadrana and Colot, 2016*; *Richards et al., 2017*; *Johannes and Schmitz, 2019*). This is largely because, in most species, it is difficult to distinguish between epigenetic variation that results from genetic differences and pure epigenetic variation, which is independent of the genotype (*Richards, 2011*). Disentangling genetic and epigenetic effects bears most promise in clonally reproducing species, such as dandelions or strawberries (*Preite et al., 2018*; *Sammarco et al., 2024*), or in highly inbred species, such as *Arabidopsis*.

In a previous experiment with *Arabidopsis* (*Schmid et al., 2018b*), we simulated selection in a rapidly changing environment. We compared phenotypic traits and epigenetic variation between populations grown for five generations under selection with their genetically identical ancestors in a common environment. Selected populations of two distinct genotypes exhibited significantly different flowering times and plant architectures when compared with their ancestral populations, and these differences persisted for at least three generations in a common environment. In parallel, we observed a reduction in both, epigenetic diversity and changes in DNA methylation patterns, some of which were associated with the observed phenotypic changes. These findings indicated that epigenetic variation was subject to selection. The genotypes used in the described experiment were recombinant inbred lines (RILs), formed by hybridizing two different accessions, followed by several generations of inbreeding. Therefore, most of the epigenetic variation in these RILs was likely retained as standing variation from the original hybridization event (*Greaves et al., 2015*) and did not accumulate during the selection experiment through spontaneously occurring or environmentally induced epimutations (*Schmid et al., 2018b*).

Here, to assess the potential for epigenetic differentiation between selection treatments, we used a similar approach using *Arabidopsis* accessions from a previous selection experiment that investigated the effects of aphid herbivores on plant defensive traits (*Züst et al., 2012*). In this selection experiment, mixtures of genetically uniform *Arabidopsis* accessions were grown for five generations in either insect-free cages (control) or cages inoculated with different aphid species or a mixture of these species. After five generations, genetic variation was significantly reduced, and populations were dominated by three accessions to various degrees (*Züst et al., 2012*). To examine whether epigenetic

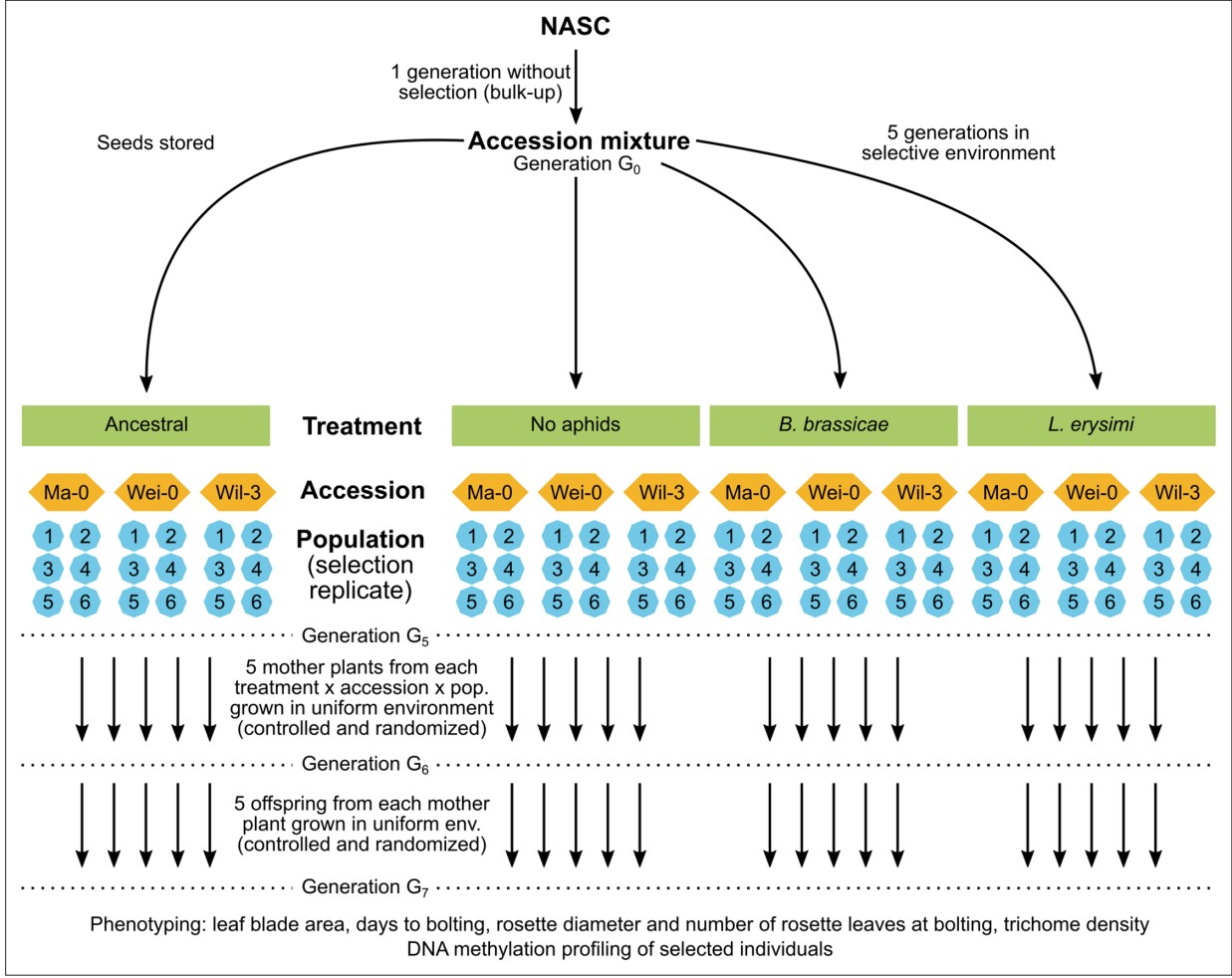

**Figure 1.** Schematic of the original experimental design (top) and the current study (bottom).

variation was also selected within these three accessions (Ma-0, Wei-0, Wil-3), we compared phenotypic traits of selected populations (i) to the original founder population, (ii) to the control population, and (iii) between two single-species aphid treatments. From each selection treatment, we used six independent replicate populations after five generations of selection and grew them together for two generations in a randomized planting pattern in a common environment. To minimize confounding maternal effects, we assessed the phenotypes only in the second generation. We further compared genome-wide DNA methylation states in a subset of mother plants used to set up the common-environment experiment (*Figure 1*).

We designed the experiment to detect effect sizes of 10% or more at the significance level of p<0.05 with a statistical power of *beta*>0.80. While we found highly significant differences between accessions, replicate populations, and across five experimental blocks (with plants generally growing worse in two blocks compared to the other three), we observed only few significant differences between selection regimes (control and the two different aphid treatments). These differences were weak, with the percentage sum of squares explained by the contrasts being less than 1% and p-values ranging between 0.05 and 0.01. We found minimal differences in DNA methylation patterns within accessions between plants from different selection regimes, with 0.009% of all tested cytosines showing significant differences after correction for multiple testing (false discovery rate [FDR]<0.05). In contrast to our previous study (*Schmid et al., 2018b*), there was no reduction in the amount of epigenetic variation in response to selection. In summary, we found evidence that some within-accession differences caused by the selection regimes persisted for two generations in the common environment. However, the extent of these differences was low. Thus, we conclude that the evidence for selection of epigenetic variation in this experiment is weak.

In strong contrast to our findings, a recent study claimed that environment-induced heritable variation is common in *Arabidopsis* (*Lin et al., 2024*). We were surprised about this finding because, in our experiment, the aphid treatments had a strong impact on plant fitness and survival but not on epigenetic differentiation. Thus, if environment-induced heritable variation was indeed so common in this model species, we would have expected to observe it, too. We noticed that the design of the study by *Lin et al., 2024* was elegant but susceptible to potential errors that confounded treatments with genetic variation at the beginning of the experiment. We used their RNA-seq data to assess whether the analyzed plants displayed any evidence of residual genetic variation that was potentially confounded with the treatments. Our analysis showed that about 10% of the plants that we could assess belonged to accessions differing from the ones they had been assigned to. This error did not occur at random but stemmed from a mix-up of two selection treatments. Thus, the large effects assigned to environment-induced heritable variation were very likely due to genetic differences between accessions.

In summary, while epialleles can be subject to selection and contribute to adaptation (*Schmid et al., 2018b*), the frequency of these events remains unclear, and epigenetic differentiation in response to selection seems to require standing epigenetic variation within accessions. In contrast, evidence for a role of environment-induced epigenetic changes in adaptation in the model plant *Arabidopsis* remains weak.

## Results

### Impact of selection treatments on phenotypic traits

As described above, mixtures of genetically uniform *Arabidopsis* accessions were grown for five generations, either under control conditions or inoculated with different aphid species (*Züst et al., 2012*). Thereafter, the populations were dominated by the three accessions Ma-0, Wei-0, and Wil-3. Phenotypic differences within uniform genetic backgrounds that were subjected to different treatments would indicate the selection of epigenetic variation (*Figure 1*). Thus, to investigate whether the treatments led to phenotypic differences within the three dominant accessions, we tested whether

**Table 1.** Percent sum of squares explained by model terms and p-values using all three accessions.

Contrasts of a term are marked with indents. Complete output and further results for individual accessions are given in *Supplementary files 1 and 2*.

| Model term | Blade area % SS | p-value | Time to bolting % SS | p-value | Rosette diameter % SS | p-value | Number of rosette leaves % SS | p-value |
|---|---|---|---|---|---|---|---|---|
| Blocks | 46.97 | <0.001 | 6.55 | <0.001 | 43.23 | <0.001 | 27.1 | <0.001 |
| >Blocks 1–3 vs. blocks 4–5 (EX) | 41.85 | <0.001 | 0.51 | <0.001 | 39.62 | <0.001 | 26.42 | <0.001 |
| >Remaining blocks | 5.12 | <0.001 | 6.04 | <0.001 | 3.61 | <0.001 | 0.68 | <0.001 |
| Selection treatments (ST) | 0.1 | 0.887 | 0.24 | 0.925 | 0.76 | 0.033 | 0.5 | 0.782 |
| >Ancestral vs. rest (AN) | 0.02 | 0.742 | 0.06 | 0.735 | 0.11 | 0.222 | 0.24 | 0.484 |
| >Control vs. treated (CO) | 0.07 | 0.495 | 0.05 | 0.757 | 0.33 | 0.043 | 0.21 | 0.506 |
| >*B. brassicae* vs. *L. erysimi* (SE) | 0.01 | 0.852 | 0.13 | 0.624 | 0.32 | 0.047 | 0.05 | 0.746 |
| Original replicate population (LI) | 2.38 | <0.001 | 8.82 | <0.001 | 1.18 | <0.001 | 7.88 | <0.001 |
| Accession (ET) | 8.16 | <0.001 | 23.23 | <0.001 | 2.2 | <0.001 | 14.7 | <0.001 |
| ET:ST | 0.42 | 0.026 | 0.34 | 0.41 | 0.62 | 0.002 | 0.16 | 0.655 |
| >ET:AN | 0.05 | 0.315 | 0.09 | 0.427 | 0.35 | 0.001 | 0.12 | 0.238 |
| >ET:CO | 0.06 | 0.288 | 0.22 | 0.153 | 0.01 | 0.84 | 0.01 | 0.843 |
| >ET:SE | 0.31 | 0.005 | 0.03 | 0.776 | 0.26 | 0.005 | 0.03 | 0.707 |
| ET:LI | 0.37 | 0.443 | 0.88 | 0.076 | 0.3 | 0.864 | 0.66 | 0.175 |
| Mothers | 6.07 | 0.007 | 8.11 | 0.216 | 7.38 | 0.024 | 6.81 | 0.342 |

**Table 2.** Percent sum of squares explained by model terms and p-values using only accessions Ma-0 and Wei-0. Contrasts of a term are marked with indents. Complete output and further results for individual accessions are given in *Supplementary files 1 and 2*.

| Model term | Blade area % SS | p-value | Time to bolting % SS | p-value | Rosette diameter % SS | p-value | Number of rosette leaves % SS | p-value | Trichome density % SS | p-value |
|---|---|---|---|---|---|---|---|---|---|---|
| Blocks | 48.54 | <0.001 | 5.91 | <0.001 | 42.45 | <0.001 | 29.24 | <0.001 | 28.97 | <0.001 |
| >Blocks 1–3 vs. blocks 4–5 (EX) | 43.45 | <0.001 | 0.28 | 0.026 | 38.73 | <0.001 | 27.94 | <0.001 | 25.28 | <0.001 |
| >Remaining blocks | 5.09 | <0.001 | 5.63 | <0.001 | 3.72 | <0.001 | 1.3 | <0.001 | 3.69 | <0.001 |
| Selection treatments (ST) | 0.06 | 0.926 | 0.53 | 0.827 | 0.19 | 0.613 | 0.19 | 0.941 | 0.18 | 0.901 |
| >Ancestral vs. rest (AN) | 0 | 0.955 | 0.17 | 0.597 | 0 | 0.888 | 0.11 | 0.641 | 0 | 0.936 |
| >Control vs. treated (CO) | 0 | 0.908 | 0.21 | 0.561 | 0.1 | 0.338 | 0.07 | 0.715 | 0.02 | 0.801 |
| >*B. brassicae* vs. *L. erysimi* (SE) | 0.06 | 0.515 | 0.15 | 0.628 | 0.09 | 0.367 | 0.01 | 0.874 | 0.16 | 0.49 |
| Original replicate population (LI) | 2.16 | <0.001 | 9.5 | <0.001 | 1.67 | 0.002 | 7.96 | <0.001 | 4.99 | <0.001 |
| Accession (ET) | 5.78 | <0.001 | 18.5 | <0.001 | 2.46 | <0.001 | 11.78 | <0.001 | 13.98 | <0.001 |
| ET:ST | 0.12 | 0.424 | 0.42 | 0.349 | 0.38 | 0.036 | 0.04 | 0.799 | 0.07 | 0.572 |
| >ET:AN | 0.02 | 0.471 | 0.19 | 0.225 | 0.13 | 0.06 | 0.02 | 0.51 | 0.04 | 0.304 |
| >ET:CO | 0.03 | 0.394 | 0.18 | 0.241 | 0.01 | 0.588 | 0.02 | 0.522 | 0.02 | 0.422 |
| >ET:SE | 0.07 | 0.222 | 0.05 | 0.515 | 0.24 | 0.017 | 0 | 0.775 | 0.01 | 0.66 |
| ET:LI | 0.34 | 0.241 | 0.89 | 0.048 | 0.21 | 0.775 | 0.3 | 0.609 | 0.28 | 0.597 |
| Mothers | 7.69 | 0.023 | 11.27 | 0.319 | 9.16 | 0.2 | 10.2 | 0.106 | 8.25 | 0.46 |

the selection treatments had an impact on the time to bolting, the rosette diameter at bolting, the number of rosette leaves at bolting, the leaf blade area, and the trichome density (*Tables 1 and 2*; *Figures 2 and 3*). Besides the selection treatment and its contrasts, the model also included the experimental blocks (n=5), the accessions (n=3), the populations (n=18 + 3 ancestrals), and maternal lines ('mother', n=261), along with interactions of the selection treatment contrasts with accessions. Additionally, we conducted analyses without the Wil-3 accession because this accession had low replication at the level of maternal lines. We also ran separate analyses for each accession. Additional models and power estimations can be found in *Supplementary files 1 and 2*.

The term 'block', especially the comparison between blocks 1–3 and blocks 4–5, was highly significant in all analyses and generally explained most of the total phenotypic variation (up to nearly 50% for leaf blade area, *Table 2*). The time to bolting was an exception because here the term 'accession' explained most of the variation (23.2%), followed by 'population' (8.8%), 'mother' (8.1%), and 'block' (6.5%). The terms 'accession' and 'population' were also consistently significant and explained a large fraction of phenotypic variation. Effects of 'population' were likely due to differences in composition because accessions were not balanced between populations. Accordingly, the interaction between 'accession' and 'population', which would indicate differences between populations within accessions, was rarely significant and explained less than 1% of all variance. Additionally, the term 'mother' was occasionally significant, accounting for 5–10% of the total phenotypic variance (*Tables 1 and 2*). Compared with these previous terms indicating environmental and genetic influences, the selection treatment and its contrasts – as well as their interactions with the previous terms – were rarely significant and explained less than 1% of the total phenotypic variation (*Tables 1 and 2*, *Figures 2 and 3*). Furthermore, the significant differences in rosette diameter between the control and aphid treatments and between the two aphid treatments (*Table 1*) disappeared when we excluded the accession Wil-3 (with low replication at the level of maternal lines) from the analysis (*Table 2*). In this case, only the interaction between the two aphid treatments and the two accessions Ma-0 and Wei-0 remained significant (*Table 2*). Separate analyses for each accession revealed significant differences in bolting time between the control and the aphid treatments in Ma-0 (*Figures 2 and 3*). We also observed a significant

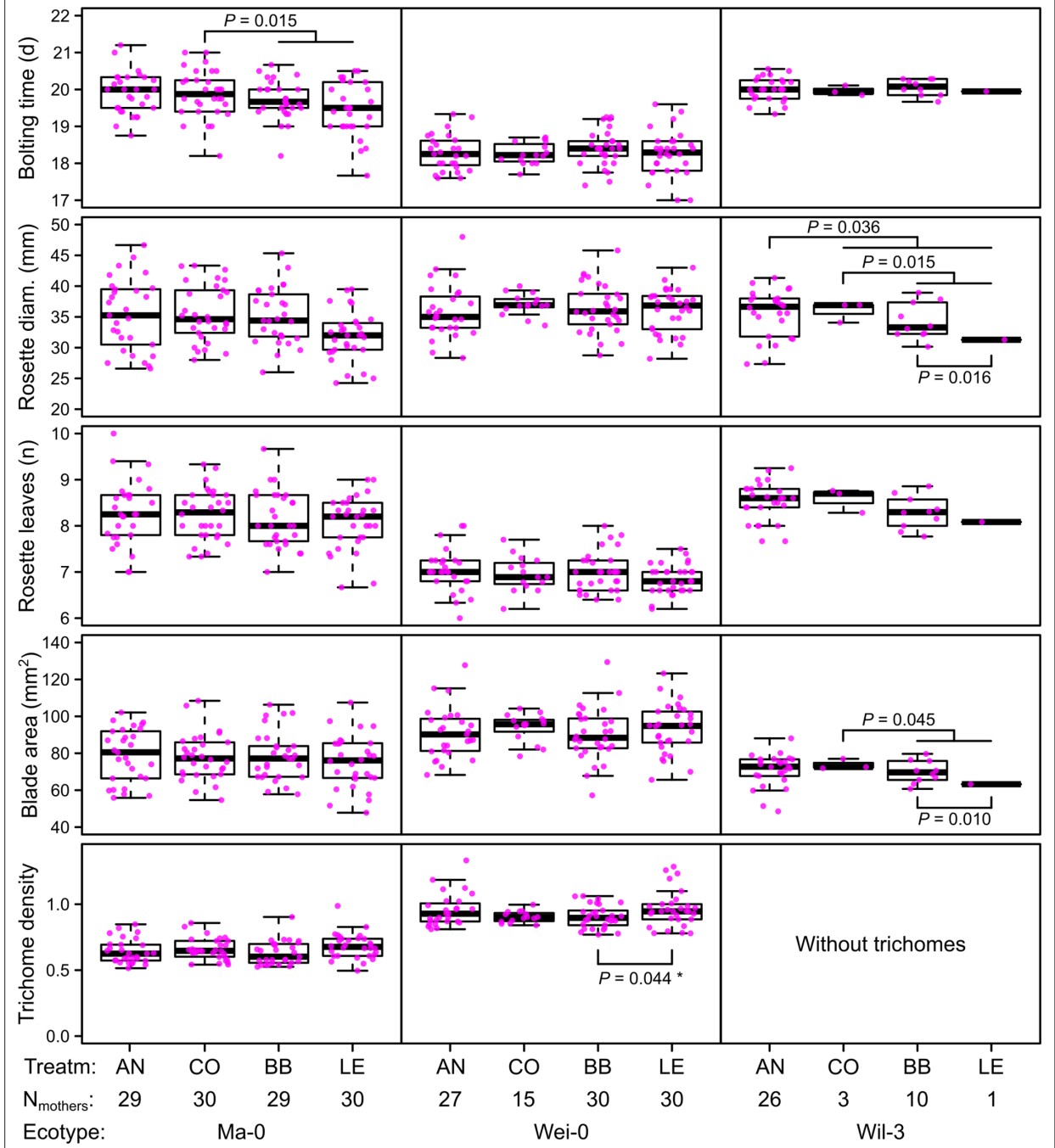

**Figure 2.** Phenotypes measured in the G7 generation averaged by the ID of the maternal lines from the G6 generation. p-values for contrasts with p<0.05 are indicated (based on the analyses done per accession). p-values with asterisk were not significant when outliers were removed (see **Figure 3**). Boxplots were drawn with the function graphics::boxplot() in R. The box extends from the 25th to the 75th percentile. The wiskers extend to the most extreme data point which is no more than 1.5 times the interquartile range from the box.

difference in trichome densities between the two aphid treatments in accession Wei-0; however, these were driven by some extreme values (**Figure 3**). We found several significant differences between selection treatments in accession Wil-3 (**Figures 2 and 3**) but, again, considering the low level of replication at the level of maternal lines in this accession, these differences should be treated with caution.

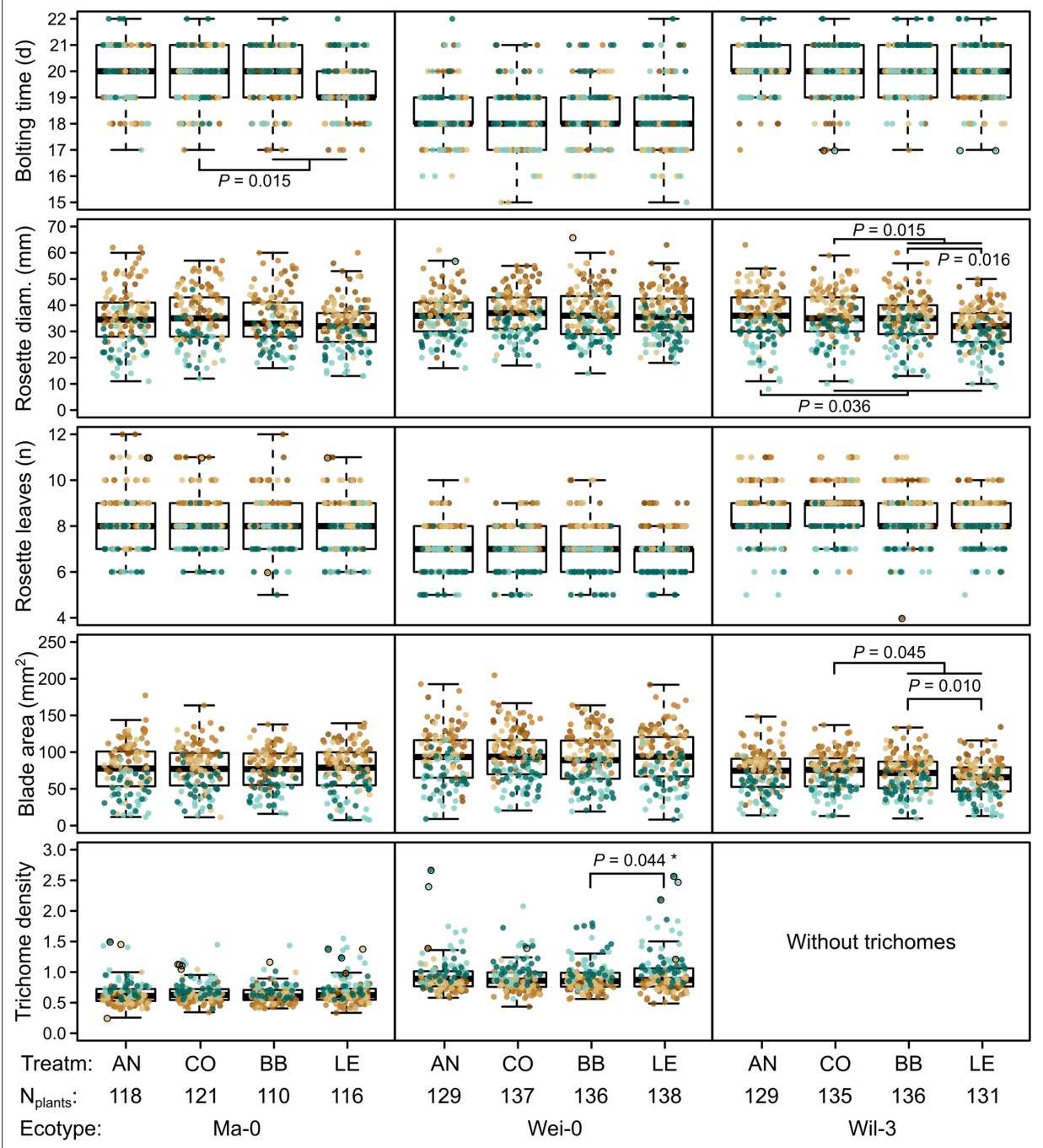

**Figure 3.** Phenotypes measured in the G7 generation. Brown and green colors represent plants from blocks 1–3 and 4–5, respectively (slower growth in blocks 4–5). Dots with black outlines were identified as outliers. The number of plants available for the test differed slightly between the measurements. The number of plants given at the bottom of the figure corresponds to the minimal number of plants (i.e., for each phenotype there were at least $N_{plants}$ plants). It varies by 2–3 plants per accession and treatment as some measures were missing. p-values for contrasts with p<0.05 are indicated (based on the analyses done per accession). p-values with asterisks were not significant when outliers were removed. Boxplots were drawn with the function graphics::boxplot() in R. The box extends from the 25th to the 75th percentile. The wiskers extend to the most extreme data point which is no more than 1.5 times the interquartile range from the box.

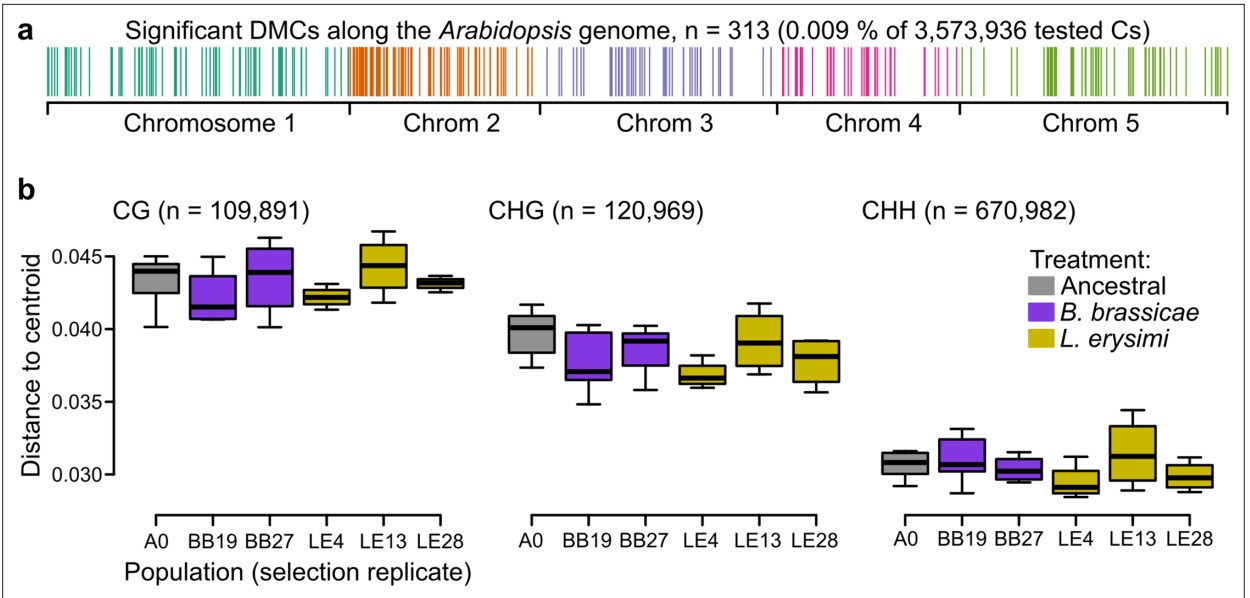

**Figure 4.** Epigenetic differences between ancestral and selected populations using a subset of mothers from the accession Wei-0. (**a**) Location of the differentially methylated cytosines along the genome. (**b**) Epigenetic variation within different populations (selection replicates), i.e., distances to each centroid. Only populations with at least four individuals are shown. Number of individuals per population: A0: 7, BB19: 5, BB27: 4, LE4: 4, LE13: 4, LE28: 4 (not shown: BB14 and BB16 with only two individuals each). No overall and no pairwise p-value was smaller than 0.05. Boxplots were drawn with the function graphics::boxplot() in R. The box extends from the 25th to the 75th percentile. The wiskers extend to the most extreme data point which is no more than 1.5 times the interquartile range from the box.

## Limited impact of aphid treatments on DNA methylation patterns and epigenetic variation

To more directly assess whether epigenetic variation could be responsible for the phenotypic differences observed between ancestral and selection treatments, we characterized changes in the genome-wide methylomes, taking DNA methylation as a proxy for epigenetic variation. We investigated the methylome of a subset of Wei-0 maternal lines from generation G6 (n=34), providing a more direct readout of the aphid treatment than would be obtained with plants from generation G7. We assessed the DNA methylation patterns of these plants by whole-genome bisulfite sequencing. In total, we tested 3,573,936 cytosines for differential methylation between ancestrals and the two aphid-selection treatments. The number of significantly differentially methylated cytosines was similar in all comparisons: it ranged from 101, when the two aphid treatments were compared, to 131, when the *L. erysimi* treatment was compared with the ancestrals. Overall, about 0.009% of all tested cytosines showed significant differences, and they were dispersed across the entire genome (*Figure 4a*). We also tested whether the variation of DNA methylation, that is the extent of epigenetic variation, was reduced by the selection treatments, an effect of selection that we had previously observed in a similar experiment (*Schmid et al., 2018b*). However, we did not find any evidence for a reduction nor an expansion of variation in DNA methylation between ancestrals and the two aphid-selection treatments in any of the three contexts of DNA methylation: CG, CHG (H being A, T, or C), and CHH (*Figure 4b*).

To understand the potential effects of differential cytosine methylation, we characterized all 308 differentially methylated cytosines (DMCs) with respect to sequence context and position relative to genes. The majority of DMCs was in the CG context (87.6%), and only a few were found in the CHG and CHH contexts (6.2% each). Considering all cytosines of the *Arabidopsis* genome, about 13%, 15%, and 72% are found in the CG, CHG, and CHH contexts, respectively (*Schmid et al., 2018b*). Thus, there was a clear enrichment of DMCs in the CG context, similar to our finding in the previous selection experiment (*Schmid et al., 2018b*) and consistent with the fact that methylation in the CG context shows the highest variability (*Kartal et al., 2020*). Almost all DMCs were located within or less than 2 kb away from genes (87.3%) or genes and transposable elements (98.1%). Together with the clear enrichment of the CG context, this suggests that most DMCs were related to differences in

**Table 3.** Gene Ontology (GO) enrichment among genes mapped by differentially methylated cytosines (DMCs).
'Found' indicates the number of different genes with the term that were mapped by DMCs. 'Expected' refers to the number of genes that were expected to be mapped by DMCs if six DMCs were randomly distributed.

| GO.ID | Term | Found | Expected | p-value |
|-------|------|-------|----------|---------|
| GO:0070646 | Protein modification by small protein removal | 7 | 1.53 | 0.00083 |
| GO:0016571 | Histone methylation | 10 | 3.07 | 0.00101 |
| GO:0008284 | Positive regulation of cell proliferation | 5 | 0.8 | 0.00119 |
| GO:0042732 | D-xylose metabolic process | 3 | 0.3 | 0.00321 |
| GO:0006487 | Protein N-linked glycosylation | 5 | 1.03 | 0.0036 |
| GO:0071702 | Organic substance transport | 25 | 11.89 | 0.00423 |
| GO:0015691 | Cadmium ion transport | 2 | 0.11 | 0.00455 |
| GO:0009834 | Plant-type secondary cell wall biogenesis | 3 | 0.35 | 0.00482 |
| GO:0006306 | DNA methylation | 7 | 2.09 | 0.00492 |
| GO:0006342 | Chromatin silencing | 8 | 2.86 | 0.00794 |
| GO:0071705 | Nitrogen compound transport | 23 | 11.74 | 0.00931 |
| GO:0050665 | Hydrogen peroxide biosynthetic process | 4 | 0.83 | 0.00945 |
| GO:1905039 | Carboxylic acid transmembrane transport | 2 | 0.15 | 0.00945 |

gene body methylation (gbM; *Bewick and Schmitz, 2017*, *Muyle et al., 2022*). We tested whether genes with certain functions were enriched for DMCs using gene ontology (GO) enrichment analysis (*Table 3*). There was an enrichment of several terms related to epigenetic regulation: 'histone methylation', 'DNA methylation', and 'chromatin silencing'. The remaining terms could be associated with transport and growth in general, for example, 'positive regulation of cell proliferation', 'secondary cell wall biogenesis', and 'organic substance transport'. In conclusion, although we identified only a few DMCs, they were preferentially found in genes associated with epigenetic regulation and transport and growth processes. However, the overall limited differentiation and the absence of reduced variation in DNA methylation suggest that there was only weak selection of specific DNA methylation sites.

## The large effects observed by *Lin et al., 2024* are not due to environment-induced heritable variation but due to differences in genotype

Compared with our study, which provided weak evidence for the selection of epigenetic variation, a recent report claimed that environment-induced heritable variation is common in *Arabidopsis* (*Lin et al., 2024*). We were surprised by this difference in the outcomes of the two experiments, which both aimed at investigating the potential role of epigenetic variation in adaptation in this model species. Our aphid treatments strongly impacted plant fitness (*Züst et al., 2012*), indicating that an induction of heritable epigenetic variation within genetically uniform *Arabidopsis* populations, as suggested by *Lin et al., 2024*, was possible.

We wondered if issues with the experimental design may have led to the strong effects in the study of *Lin et al., 2024*. Using their RNA-seq raw data of 190 out of a total of 4,032 plants, we searched for residual genetic variation within accessions. Their RNA-seq data supposedly included four accessions. We compared pairwise genetic distances and identified the accessions most closely related to the experimental plants. We used 10,000 randomly drawn genes for the analyses and were left with up to 13,418 variants after filtering for coverage between 10 and 1000 in at least four plants per treatment and accession. Visualizing pairwise genetic distances between all plants revealed that two presumed accessions, Abd-0 and TRE-1, were each represented by two genetically distinct clusters (*Figure 5a*). When we visualized each accession separately, highlighting the treatments, we found that the genetically distinct groups had been associated with different selection treatments, leading to an unwanted confounding of selection treatments with accessions. Specifically, in the high Cd treatment, the plants

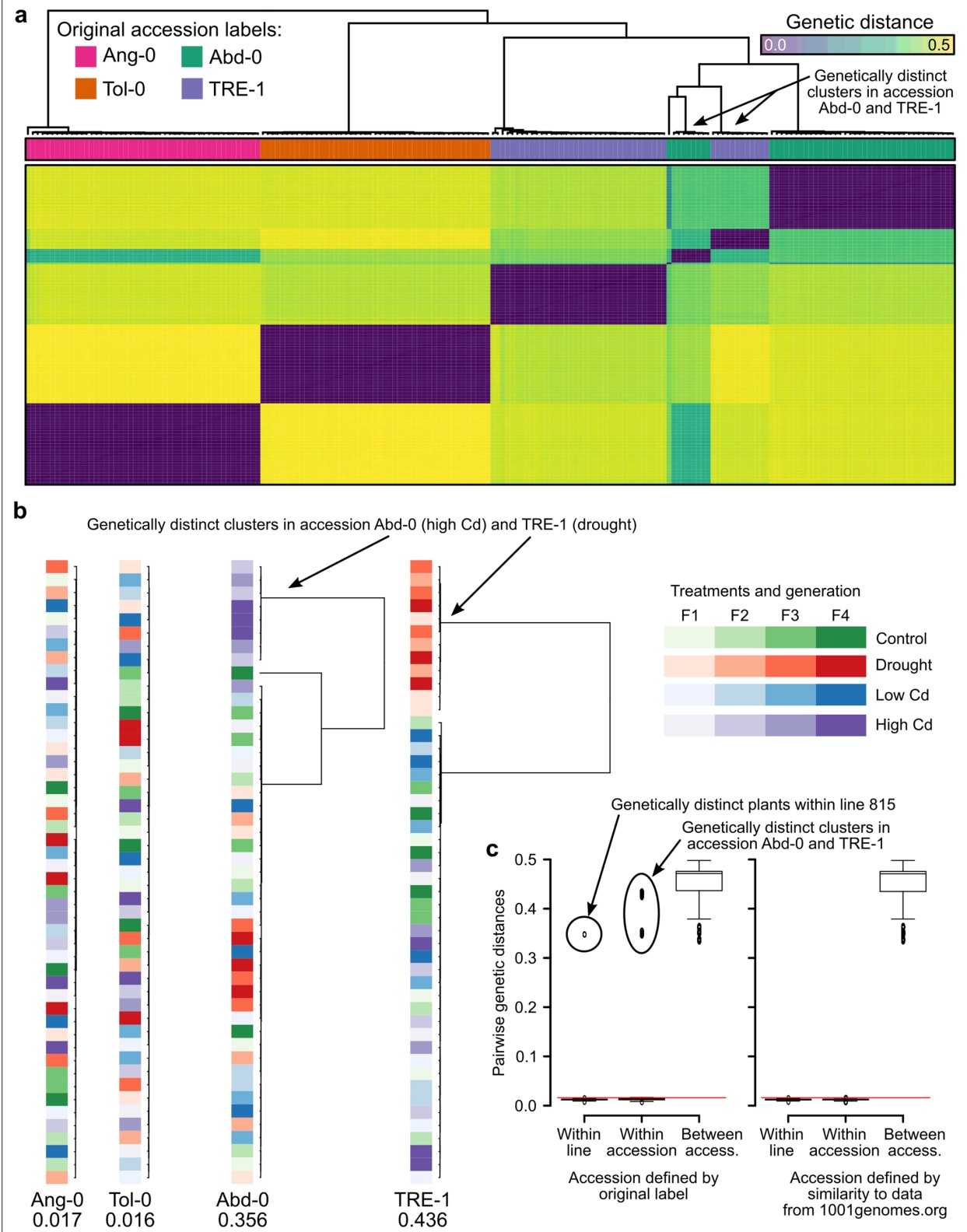

**Figure 5.** Genetic differentiation between some treatments in the study of **Lin et al., 2024**. (**a**) Genetic distances between the 190 RNA-seq samples from **Lin et al., 2024**. The two genetic clusters diverging from the rest of their accession group are highlighted. (**b**) Within accession-label genetic distances, drawn to scale. Maximal distance given below the label, colored according to treatment and generation. The two treatments that have been assigned the wrong accessions are highlighted. (**c**) Pairwise genetic distances within lines (plants of the same lineage, i.e., single-seed descendants),

*Figure 5 continued on next page*

*Figure 5 continued*

within accessions, and between accessions using either the original accession labels from *Lin et al., 2024* (left; 26) or the accessions assigned based on genetic similarity to the data available at 1001genomes.org (right). The red line demarcates the distance equal to the average of within-line distances plus three standard deviations. Boxplots were drawn with the function graphics::boxplot() in R. The box extends from the 25th to the 75th percentile. The wiskers extend to the most extreme data point which is no more than 1.5 times the interquartile range from the box.

labeled as Abd-0 were in fact from another accession, and the same happened in the drought treatment where plants labeled as TRE-1 came from another accession (*Figure 5b*). When we compared the genomes of these experimental plants to those of publicly available accessions, we found that the genetically distinct plants from the high Cd treatment belonged to the accession Ws-2 and those from the drought treatment belonged to the accession Fei-0, both accessions that were otherwise included in the experiment (*Supplementary file 3*). Notably, when we assigned the experimental plants to publicly available accessions, the distances to the closest accessions were between 0.0003 and 0.0034 (0.0019 on average), compared with distances to accessions Abd-0 and TRE-1, respectively, ranging from 0.358 to 0.451 (0.414 on average).

To confirm our finding of a potential mix-up leading to treatment–accession confounding, we used either the original or the newly assigned accession labels to split the pairwise distances between experimental plants into three classes: (i) within lineage (i.e., [grand]-parent-to-offspring), (ii) within accessions, and (iii) between accessions. The former two should both yield values close to zero as plants within lineages and within accessions should be genetically (nearly) identical to each other, whereas distances between accessions can be substantial (*Figure 5c*). These expectations were not met with the original accession labels; instead, we found large distances in all three classes. In contrast, the same evaluation with the re-assigned accession labels Ws-2 (for Abd-0 in high Cd treatment) and Fei-0 (for TRE-1 in drought treatment) confirmed the expectations of close to zero distances in classes (i) and (ii), and large distances in class (iii). Thus, it appears that *Lin et al., 2024* inadvertently had mixed up accessions when assigning plants to the two mentioned treatments.

In summary, our analysis reassigned 20 out of 190 plants (~10.5%) to new accessions. This error did not occur at random but stemmed from the wrong treatment assignment of plants from two accessions. This misassignment explains the large effects interpreted as environment-induced heritable variation by *Lin et al., 2024* that, as we show here, are really effects of genetic differences between accessions.

## Discussion

In this study, we investigated whether selection by different aphid species within genetically uniform *Arabidopsis* accessions can lead to heritable phenotypic variation that is stable in the absence of selection for at least two generations, indicative of adaptation. Such (meta)stable phenotypic variation is expected to be predominantly epigenetic in nature, defined as a mitotically and/or meiotically heritable change in gene expression without a change in DNA sequence (*Russo et al., 1996*). As *Arabidopsis* is highly inbreeding, genetic variation within an accession is expected to be rare, while epigenetic differences, for example, in DNA methylation, are more common (*Becker et al., 2011*; *Schmitz et al., 2011*). We investigated epigenetic variation at multiple levels: differences between selection treatments within accessions would indicate epigenetic differentiation, while differences between replicate populations of these accessions could arise from either standing (epi)genetic variation at the beginning of the selection experiment or from de novo gained (epi)genetic variation developed during the experiment, with genetic variants expected to be extremely rare in the case of *Arabidopsis*.

In our analyses, we found only weak evidence for epigenetic differentiation due to selection treatments, expected to manifest as significant phenotypic differences in the second generation in a common environment or as clear differences in the methylomes analyzed in our experiment. Except for one case, the differences were either limited to one accession with low replication at the level of mother plants or depended on the presence of a few extreme measurements. While differences between experimental blocks often explained the largest amount of the observed variation in phenotypes, this was different in the case of 'time to bolting'. This finding is in line with the notion that flowering time in *Arabidopsis* has a strong genetic component (*Atwell et al., 2010*). Similarly,

we consistently observed strong differences between the accessions, again emphasizing the genetic component of trait variability in this model species.

While we found little evidence for differences between populations within accessions, we found significant differences between maternal lines. In genetically uniform accessions, there are three possible explanations for these maternal line effects: (i) maternal effects, (ii) initial genetic or epigenetic differences between maternal lines that resulted in founder effects, and/or (iii) a differential (but selection treatment-independent) gain of genetic or epigenetic variation between maternal lines during the five generations of the selection experiment.

Initial genetic (or epigenetic) differences between maternal lines (or populations) may have two reasons. First, seeds of single accessions obtained from NASC may not have been derived from homozygous, genetically uniform plants. Second, there could have been genotyping errors occurring during the identification of plants representing the accessions of interest. The latter case applied to the study of *Lin et al., 2024*, in which two presumed accessions, Abd-0 and TRE-1, were thought to show strong evidence of environment-induced heritable variation while the other two, Ang-0 and Tol-0, did not. Plants of the two accessions showing no effect were genetically very similar to each other and were found identical to their accessions using publicly available data from over 1000 accessions (1001genomes.org). However, plants from the presumed accessions with strong effects, Abd-0 and TRE-1, showed strong genetic differences between treatments, which results from a mix-up of accessions while preparing or conducting the experiment.

While experimental errors can only be addressed by verifying the material's identity at each step (which can be costly and challenging), residual genetic variation and newly acquired (epi)genetic variation could be experimentally distinguished in certain setups. If plants of single accessions are split up into populations that are maintained throughout the experiment, it is important to grow and phenotype them in a common environment both at the beginning and at the end of the experiment. If phenotypic differences between replicate populations of single accessions already occurred before imposing the selection treatments, it would suggest that the populations were (epi)genetically distinct from each other. However, if phenotypic differences between replicate populations emerge after the treatment and are maintained in a common environment, then this indicates that variation was gained during the treatment. We note that differences between replicate populations due to founder effects or drift may be as relevant regarding evolutionary processes as are consistent differences between selection treatments (*Crow and Kimura, 1970*).

We found weaker evidence for selection of epigenetic variation in this *Arabidopsis* experiment than we did in our previous study, in which we experimentally simulated selection in a rapidly changing environment (*Schmid et al., 2018b*). The reason for this difference may be that, in the previous study, we used RILs that were generated by crossing two different accessions, followed by several generations of inbreeding. We hypothesized that the epialleles differing between selected and ancestral plants persisted for multiple generations in a state that did not match the underlying genotype but finally converged to it (*Schmid et al., 2018b*). This scenario was impossible in the present study because we used distinct accessions. In our previous study (*Schmid et al., 2018b*), we also noted some residual heterozygosity in one of the RILs, which might have contributed to our observations. Studies on a wide range of plant species that included genetically non-uniform populations consistently reported a significant correlation between genetic and epigenetic variation (*van Moorsel et al., 2019*; *Mounger et al., 2021*; *Boquete et al., 2022*; *Mounger et al., 2022*; *Ibañez et al., 2023*). To the best of our knowledge, there are very few studies in natural systems that demonstrate genetically independent, heritable – that is, 'true' epigenetic – variation in the absence of environmental triggers or after accounting for maternal effects. For example, maternal effects may explain the differences in flowering time between genetically identical apomictic dandelions collected in the field (*Wilschut et al., 2016*). One example of inheritance of true epigenetic variation has recently been shown in the clonally reproducing duckweed *Lemna minor* (*van Antro et al., 2023*). The authors report an epigenetic memory effect of heat treatments with changes in DNA methylation in the CHG context. However, this might be due to differences in mitotic (i.e., clonal) vs. meiotic inheritance. While it has long been known that certain epigenetic states are faithfully inherited during mitosis (*Meins, 1983*; *Meins, 1986*), recent evidence shows that meiosis is accompanied by epigenetic reprogramming in plants (*Slotkin et al., 2009*; *She and Baroux, 2014*; *She and Baroux, 2015*; *Park et al., 2016*; *Schmid et al., 2018a*; *Borg et al., 2021*). Thus, it may be interesting to study epigenetic inheritance in species

that generally reproduce clonally through mitosis but distinguish between vegetative reproduction through multicellular meristems and apomictic reproduction through an unreduced, unrecombined egg cell (*Schmid, 1990*; *Koltunow and Grossniklaus, 2003*; *Schmidt et al., 2015*).

## Materials and methods

### Experimental background and material from a multigenerational selection experiment

The selection experiment from which the plants used in this study were derived was fully described by *Züst et al., 2012*. In brief, the experiment began with a collection of 27 *Arabidopsis* accessions. Seeds obtained from NASC were expected to represent genetically uniform accessions and were propagated for one generation. For the selection treatment, experimental populations with 20 seeds from each accession were planted in cages. 17 days after sowing, different aphid species or a mixture were released into the cages (none as a control). After 60 days, all plants in a cage were harvested and a population of 800 seeds was sown representing the next generation. Each population was maintained over five generations of selection, with six replicate populations per selection treatment. At the end of the selection experiment, the genetic composition of the populations was assessed (*Züst et al., 2012*).

### Experimental design

For the study in a common environment, we focused on three well-surviving accessions from the selection experiment: Ma-0, Wei-0, and Wil-3. The accessions Ma-0 and Wei-0 were chosen because they were the most frequent after the aphid selection treatments. Wil-3 was chosen because it is a trichome-less accession that persisted across all treatments. We further focused on three selection treatments: the two single-aphid treatments with *Bravicoryne brassicae* (BB) and *Liphaphis erysimi* (LE), and the control without aphid treatment. Additionally, we included offspring from the plants that were used to set up the original selection experiment, here referred to as 'ancestrals'. All selected plants were grown together for one generation to minimize confounding maternal effects. For each treatment and accession, we planned to use five mother plants from each of the six populations per selection treatment. However, due to the rarity of some accessions in certain populations, we supplemented with mothers from other populations of the same selection treatment to fill up missing plants (360 maternal seed families were planned but ultimately only 261 were used). We accounted for this in our analysis by using population identity (n=6 replicates × 3 selection treatments + ancestrals = 19) and maternal plant identity (n=261 < 5 mothers × 3 accessions × 18 populations for control and selection-treatment plants + 30 mothers × 3 accessions for ancestrals = 360) as random terms. The ancestrals did not stem from multiple populations, but we planned to use an equal amount of mother plants per accession (n=30). After the first generation of the present experiment (the sixth generation [G6] in reference to the selection experiment), between 5 (planned) and 150 (if there was only a single mother available) offspring of each mother were grown and phenotyped in the second (G7) generation (n=1,800). Both generations were grown in a common environment in a fully randomized block design. Hence, if there were differences between selection treatments, they would have been maintained for two generations in the absence of selection by aphids.

### G6: Genotyping of offspring from the selection experiment

In the selection experiment, seeds were collected from all plants per cage (=population) at once. Thus, the accessions were mixed within populations. To identify the desired accessions, 2,000 seeds were sterilized and sown on Murashige and Skoog (MS) agar plates and stratified for 3 days at 4°C. Plates were transferred to an incubator with a day/night cycle of 18 h/6 h at a constant temperature of 22°C. Seven days after germination, we harvested leaves from seedlings resembling the Ma-0, Wei-0, or Wil-3 accession. Seedlings were genotyped as described previously (*Züst et al., 2012*). Genotyped seedlings were transplanted onto soil 14 days after germination and were grown in a controlled climate chamber at 18°C under a day/night cycle of 16 h/8 h. Seeds of plants from this G6 generation were collected from each plant separately to start the G7 generation with one seed per plant (n=1800; 1759 surviving).

## G7: Genotyping and phenotyping

The plants of the G7 generation were arranged in a randomized layout on MS agar plates and later in trays (24 plants per tray). We changed from square plates to circular plates after the first three blocks. This caused a change in the thickness of the MS agar on the plates and subsequently to slower growth in blocks 4 and 5 than in blocks 1–3. We accounted for these differences during statistical analyses by using a corresponding contrast term within the term blocks. Plants were transplanted onto soil 14 days after germination and grown in a controlled climate chamber at 18°C under a day/night cycle of 16 h/8 h. Plants were grown in five randomized blocks of 360 plants with offspring of each mother present in each block. Plants were not switched between blocks when transferred from agar to soil. The remaining seedling DNA, which was not used for genotyping, was stored at –20°C for later sequencing experiments.

Several traits were measured during the G7 generation (cf. *Schmid et al., 2018b*). The days to bolting were measured as an indication for flowering time. The number of rosette leaves and the rosette diameter (longest distance) were both assessed at the day of bolting. 21 days after germination, the fifth true leaf was removed and stuck onto a foil for trichome number counts and leaf blade area measurements. To count trichomes, leaves were imaged on a Leica DMR bright-field microscope. Leaf area was measured by scanning the foil with an office scanner and extracting the leaf area with the ImageJ plugin LeafJ (*Maloof et al., 2013*).

## Statistical analyses

### Statistical models

All analyses were carried out in R version 3.6.1. Raw data are available (*Supplementary file 4*). The design comprised the following factorial terms that were not of interest per se: accession (ET, 3 levels), block (BL, 5 levels with a fixed-effect contrast EX comparing blocks 1–3 with blocks 4–5), tray (TR, 75 levels), population identity (LI, 18 levels +1 ancestral per accession), mother plant identity (MO, <120 levels per accession, 261 in total). The factorial term of interest was the selection treatment (Selection, 4 levels), which we split into three orthogonal one-degree of freedom contrasts to compare (i) ancestral plants with plants from the selection experiment (AN), (ii) the control treatment with the two selection treatments (CO), and (iii) the two selection treatments with each other (SE). We used both the regular lm() function with specifying the appropriate statistical tests manually and lmer() that handles linear mixed models with random terms (*Schmid et al., 2017*). With the regular lm(), we used the following two formulae for the explanatory terms:

- (EX+BL)+TR+(AN+CO+SE+LI)+ET+(AN+CO+SE+LI):ET+MO (m1)
- (EX+BL)+TR+(AN+CO+SE+LI)+ET+EX:(AN+CO+SE+LI)+EX:ET+LI:ET+MO (m2)

The term selection and its contrasts were tested against the term population identity (LI). Interactions of selection contrasted with accession (e.g., SE:ET) or with the block contrast (e.g., SE:EX) were tested against the interaction between population identity and accession (LI:ET). The accession term itself (ET) was tested against the term mother plant identity within accessions (MO). Both models were also fitted with lmer() using the following formulae:

- (AN+CO+SE)+ET+(AN+CO+SE):ET+(1|BL)+(1|TR)+(1|MO)+(1|LI/ET) (m1)
- EX+(AN+CO+SE)+ET+EX:(AN+CO+SE)+ET:EX+(1|BL)+(1|TR)+(1|MO)+(1|LI/ET) (m2)

The two approaches yielded practically identical results, and we refer to the results obtained with the classical lm() function in the main text. Results from the lmer() tests are given in *Supplementary file 1*.

We ran the models for all phenotypic traits separately. We transformed all traits using the logarithm and the square root and tested whether the transformations improved the normality assumption for the residuals using a Shapiro–Wilk test (*Shapiro and Wilk, 1965*). In general, transformations did not improve the normality assumption except for trichome density, which was then transformed using the logarithm. We further assessed the effects of outliers by removing potential outliers with the ROUT method using the accession and the selection treatment for grouping (*Motulsky and Brown, 2006*). Accession Wil-3 had only one LE and three BB mothers, meaning that the two selection treatments were poorly replicated for this accession. As Wil-3 lacks trichomes, it was excluded from the trichome

density analysis. We also ran all the models without Wil-3. To demonstrate that we tested exhaustively, we report all our results.

## Power estimations

To estimate the statistical power to detect selection effects within genotypes (i.e., accessions), we employed the same models as described above and simulated data using the same data structure as in the real data. We simulated five different settings with effect sizes ranging from 1% to 20%: (i) differences between all four groups with step sizes of 1%, 5%, 10%, 15%, and 20% between each group (with the level following AN<CO< BB<LE, note that the largest difference here is 60%); (ii) only different in ancestors; (iii) only different in controls; (iv) different in both selection treatments; and (v) different in only one selection treatment. For each setting, we generated 1,000 data sets and employed the models as described above. We then summarized the percentage of significant outcomes for the different settings (*Supplementary file 2*). In brief, we could detect effect sizes of 10% in more than 80% of the simulated data sets. An exception was trichome density, for which the effect size had to be 20% for a robust detection. Note that an effect size of 20% is generally considered small (*Cohen, 1988*).

# DNA methylation

## Library preparation

We selected 7, 12, and 15 mother plants from the ancestral, BB, and LE treatments (accession Wei-0). Libraries for whole-genome bisulfite sequencing were prepared as previously described (*Schmid et al., 2018b*), using the DNA extracted for genotyping of the plants in the G6 generation and sequenced on an Illumina HiSeq 2500 (250 bp paired-end). In total, we obtained 293,022,365 read pairs. On average, samples had about 8.6 mio reads. The lowest were 3.8, 6.4, and 6.5 mio, the highest were 11.7, 12.8, and 13.8 mio reads. Mapping efficiencies were generally similar (between 37.8% and 48.2%) but two samples had clearly lower mapping efficiencies (6.0% and 13.4%). These two samples were removed from the analysis (A18 and B26).

## Data processing

Short reads were (quality) trimmed with fastp (version 0.20.0 with the options `--trim_front1 6 --trim_tail1 6 --trim_front2 6 --trim_tail2 6`; *Chen et al., 2018*) and aligned to the reference genome with Bowtie2 (version 2.3.5.1, *Langmead and Salzberg, 2012*) in combination with Bismark (version 0.22.3 with the option `--dovetail`, *Krueger and Andrews, 2011*). The reference genome for accession Wei-0 was obtained from 1001genomes.org (GMI-MPI release v3.1, pseudogenomes, pseudo6979.fasta.gz) and plastid sequences were added from the Col-0 reference genome (TAIR10). Alignments were deduplicated with Bismark and methylation tables were extracted with MethylDackel (version 0.5.0, github.com/dpryan79/MethylDackel, RRID:SCR_025850). Files were merged and only cytosines with a coverage between 5 and 100 within at least two samples per group were kept (434,671; 479,201; and 2,669,064 in the CG, CHG, and CHH context, respectively; in total about 8.5% of all available cytosines). The three experimental groups (ancestrals and *B. brassicae* or *L. erysimi* aphid treatments) were compared to each other with the R package DSS (version 2.32.0, *Park and Wu, 2016*), using the functions DMLfit.multiFactor(), DMLtest.multiFactor(), and callDMR(). The p-value threshold for the DMR calling was set to 0.01. Finally, all p-values were corrected for multiple testing and thereby converted to FDRs (*Benjamini and Hochberg, 1995*).

To test whether the extent of epigenetic variation changed during the experiment, we compared variation within original replicate populations with at least four samples with the function betadisper() from the R-package vegan (version 2.5.7, *Dixon, 2003*, *Mounger et al., 2021*). The function was also used to extract the distances to the centroids. From the 32 samples, 28 were within original replicate populations with at least 4 samples: 7 ancestrals, 12 from the *L. erysimi* treatment (3 original replicate populations), and 9 from the *B. brassicae* treatment (2 original replicate populations). Epigenetic distances between samples were calculated as the average difference in percent methylation across all cytosines (*van Moorsel et al., 2019*) that passed the following filter: coverage between 5 and 100 in at least 5 individuals per treatment (n=901,843).

To functionally characterize the genes associated with DMCs, we tested for enrichment of GO terms using topGO 2.20 (*Alexa et al., 2006*) in conjunction with the GO annotation available through

biomaRt (*Durinck et al., 2009*). Analysis was based on gene counts (genes with DMCs within 2 kb compared to all genes with one or more tested cytosines that had an average of at least 10% methylation, i.e., potential gbM genes) using the 'weight' algorithm with Fisher's exact test (both implemented in topGO). A term was identified as significant within a given parameter combination if the p-value was below 0.01.

## Analysis of genetic variation in the subset of 190 plants studied by *Lin et al., 2024*

To assess whether there was any genetic variation associated with the treatments in the experiment described by *Lin et al., 2024*, we used their RNA-seq data (PRJNA997595) on a subset of 190 plants that supposedly included four accessions. To obtain DNA sequence variant data from RNA-seq data, we followed, as far as possible (i.e., base recalibration required known sites that were not available), the GATK-guidelines. In brief, short reads were (quality) trimmed with fastp (version 0.23.4; *Chen et al., 2018*) and aligned to the reference genome with STAR (version 2.7.11a with the options `--alignIntronMax 10000 --alignMatesGapMax 10000 --outSAMstrand-Field intronMotif` and supplying the sample ID as read-group information; *Dobin et al., 2013*). Only unique alignments with a quality of at least 250 were kept. Read duplicates were marked with Picard tools (version 2.18.25; broadinstitute.github.io/picard; RRID:SCR_006525) and spliced reads were split with GATK SplitNCigarReads. Bam files were merged into a single file to reduce running time while calling variants. Data were subset to 10,000 randomly chosen genes. Variants were finally called with GATK HaplotypeCaller (with the options `--dont-use-soft-clipped-bases --standard-min-confidence-threshold-for-calling 20.0`) and filtered with GATK VariantFiltration (with the options `--window 35 --cluster 3 --filter-name "FS"--filter "FS>30.0" --filter-name "QD" --filter "QD<2.0"`). The resulting VCF file was then filtered for variants with a coverage between 10 and 1,000 within at least four plants per accession and treatment. Variants with a minor allele frequency below 5% were removed. 13,418 variants passed this filter. We did not filter accessions separately when plotting them individually to maintain the distance values. Genetic pairwise distances were calculated as the fraction of alleles that differed between two individuals (*Schmid et al., 2024*). We visualized the distances as previously described (*Schmid et al., 2024*). When we observed that some plants within two of the accessions were clearly distinct from each other, we used the data on more than 1000 *Arabidopsis* accessions available from 1001genomes.org (https://1001genomes.org/data/GMI-MPI/releases/v3.1/1001genomes_snp-short-indel_only_ACGTN.vcf.gz). We intersected the two data sets (12,686 variants remained) and calculated all pairwise distances between the experimental plants and the publicly available accessions. Experimental plants were then assigned to the accession that was closest to it (*Supplementary file 3*).

## Acknowledgements

We thank Tobias Züst (University of Zurich) for providing the seeds which served as starting material for this study, Christian Heichinger (Roche) for advice on genotyping and experimental setup, and Christina Westermann (University of Zurich) for editing the manuscript; Vimal Rawat, Gregor Rot, and Deepak Kumar Tanwar (University of Zurich) for discussions; Valeria Gagliardini, Christof Eichenberger, Daniela Guthörl, Arturo Bolaños, and Peter Kopf (University of Zurich) for general lab support; and Karl Huwiler (University of Zurich) for plant care. This work was supported through core funding of the University of Zurich and a project of the University Research Priority Program 'Evolution in Action'.

## Additional information

### Competing interests

Marc W Schmid: is affiliated with MWSchmid GmbH. Bernhard Schmid: Reviewing editor, eLife. The other authors declare that no competing interests exist.

## Funding

| Funder | Grant reference number | Author |
|---|---|---|
| University of Zurich | | Ueli Grossniklaus |
| University Research Priority Program "Evolution in Action" | | Bernhard Schmid<br>Ueli Grossniklaus |

The funders had no role in study design, data collection and interpretation, or the decision to submit the work for publication.

## Author contributions

Marc W Schmid, Data curation, Software, Formal analysis, Validation, Visualization, Methodology, Writing – original draft, Writing – review and editing; Klara Kropivšek, Formal analysis, Validation, Investigation, Methodology, Writing – original draft; Samuel E Wuest, Formal analysis, Methodology, Writing – review and editing; Bernhard Schmid, Conceptualization, Formal analysis, Supervision, Funding acquisition, Methodology, Writing – review and editing; Ueli Grossniklaus, Conceptualization, Resources, Supervision, Funding acquisition, Validation, Methodology, Writing – original draft, Project administration, Writing – review and editing

## Author ORCIDs

Marc W Schmid ⓘ https://orcid.org/0000-0001-9554-5318
Klara Kropivšek ⓘ https://orcid.org/0000-0003-1866-4094
Samuel E Wuest ⓘ https://orcid.org/0000-0003-3982-0770
Bernhard Schmid ⓘ https://orcid.org/0000-0002-8430-3214
Ueli Grossniklaus ⓘ https://orcid.org/0000-0002-0522-8974

Reviewer #1 (Public review): https://doi.org/10.7554/eLife.106930.3.sa1
Reviewer #2 (Public review): https://doi.org/10.7554/eLife.106930.3.sa2
Author response https://doi.org/10.7554/eLife.106930.3.sa3

# Additional files

## Supplementary files

Supplementary file 1. Collection of workbooks with results from the different analyses. The files are named as <phenotype > _<data set>.xlsx. For each phenotype, there are analyses with all three accessions ('all'), only the two accessions with trichomes ('NoWil3'), or with only one accession ('OnlyWei0', 'OnlyMa0', and 'OnlyWil3'). For each of them, there is one version with all data included and one in which the extreme values were removed ('.outliersRemoved'). For example: bladeArea_all.outliersRemoved.xlsx refers to the analysis of the blade area using all three accessions and removing extreme values. Within each workbook, there are three sheets for untransformed data ('y'), log-transformed data ('logy'), and square-root-transformed data ('sqrty'). Within each sheet, the output of the two models tested either with the regular linear model (lm) or the random effects model (lmer) is given. In case of the regular lm output, alternative *F*- and p-values (F_altern and P_altern) refer to the tests against the correct error stratum (i.e., nested model). The columns F and P contain the *F*- and p-values from tests against the residuals. Regarding the model terms: '__Blocks' is the five-level factor of experimental blocks that was split into two contrasts ('EX' and 'BL'); '__Selection' is the four-level factor with the selection treatments that was split into three contrasts ('AN', 'CO', and 'SE'). '__Selection_X_accession/experiment' is accordingly the interaction of the four-level factor with the selection treatments and the accession ('ET') or the 'EX' contrast. Abbreviations: Df: degrees-of-freedom, SS: sum of squares, MS: mean squares, F: *F*-value, P: p-value, F_altern: *F*-value of nested model, P_altern: p-value of nested model, perc_SS: % sum of squares explained by the term.

Supplementary file 2. A workbook containing tables from the power estimations. For each phenotype, power was estimated using all three accessions or only the two accessions with trichomes. In case of trichome density, only the latter is available. Each table contains the percentage of significant outcomes for a given simulation and factor/contrast of interest (based on 1,000 simulations). The simulations are named as <type > _<effect size>. There were five different

types of simulations with effect sizes ranging from 1% to 20%: ('step') differences between all four groups with step sizes of 1%, 5%, 10%, 15%, and 20% between each group (with the level following AN<CO<BB<LE, note that the largest difference here is 60%); ('onlyAn') only different in ancestors; ('onlyCo') only different in controls; ('onlySe') different in both selection treatments; ('onlyLe') different in only one selection treatment.

Supplementary file 3. Table with the samples from *Lin et al., 2024*. Given are the sample IDs, line ID, treatment, generation, accession label (used by *Lin et al., 2024*), and the accession assignments based on the genotypes inferred from the RNA-seq data and the publicly available data from 1001 *Arabidopsis* accessions (this study).

Supplementary file 4. Table with the phenotype data. Rosette diameter in mm, blade area in mm². Factors are labeled as described in 'Materials and methods'.

MDAR checklist

## Data availability

Accession codes: Short reads were deposited at the NCBI Sequence Read Archive (SRA) and are available through accession number PRJNA1103971. Phenotype data were deposited on Zenodo.

The following datasets were generated:

| Author(s) | Year | Dataset title | Dataset URL | Database and Identifier |
|---|---|---|---|---|
| Schmid MW, Kropivšek K, Wuest SE, Schmid B, Grossniklaus U | 2025 | Data and scripts from: Weak evidence for heritable changes in response to selection by aphids in Arabidopsis accessions | https://doi.org/10.5281/zenodo.16017957 | Zenodo, 10.5281/zenodo.16017957 |
| Schmid MW, Kropivšek K, Wuest SE, Schmid B, Grossniklaus U | 2025 | DNA methylation patter in Arabidopsis thaliana Wei-0 | https://www.ncbi.nlm.nih.gov/bioproject/PRJNA1103971 | NCBI BioProject, PRJNA1103971 |

The following previously published dataset was used:

| Author(s) | Year | Dataset title | Dataset URL | Database and Identifier |
|---|---|---|---|---|
| Lin X, Yin J, Wang Y, Yao J, Qq Li, Latzel V, Bossdorf O, Zhang Y-Y | 2024 | RNA-seq of *Arabidopsis thaliana* Leaf | https://www.ncbi.nlm.nih.gov/bioproject/PRJNA997595 | NCBI BioProject, PRJNA997595 |

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
