## [Editor Report · eLife Assessment]

This article examines selection on induced epigenetic variation (‘Lamarckian evolution’) in response to herbivory in *Arabidopsis thaliana*. The authors find weak evidence for such adaptation, which contrasts with a recently published study that reported extensive heritable variation induced by the environment. The authors **convincingly** demonstrate that the findings of the previous study were confounded by mix-ups of genetically distinct material, so that standing genetic variation was mistaken for acquired (epigenetic) variation. Given the controversy surrounding the influence of heritable epigenetic variation on phenotypic variation and adaptation, this study is an **important**, clarifying contribution; it serves as a timely reminder that sequence-based verification of genetic material should be prioritized when either genetic identity or divergence is of importance to the conclusions.

---

## [Referee Report · Reviewer #1 (Public review)]

Summary:

The authors extended a previous study of selective response to herbivory in Arabidopsis, in order to look specifically for selection on induced epigenetic variation ("Lamarckian evolution"). They found no evidence. In addition, the re-examined result from a previously published study arguing that environmentally induced epigenetic variation was common, and found that these findings were almost certainly artifactual.

Strengths:

The paper is very clearly written, there is no hype, and the methods used are state-of-the-art.

Weaknesses:

The result is negative, so the best you can do is put an upper bound on any effects.

Significance:

Claims about epigenetic inheritance and Lamarckian evolution continue to be made based on very shaky evidence. Convincing negative results are therefore important. In addition, the study presents results that, to this reviewer, suggest that the 2024 paper by Lin et al. [26] should probably be retracted.

---

## [Referee Report · Reviewer #2 (Public review)]

In this paper, the authors examine the extent to which epigenetic variation acquired during a selection treatment (as opposed to standing epigenetic variation) can contribute to adaptation in Arabidopsis. They find weak evidence for such adaptation and few differences in DNA methylation between experimental groups, which contrasts with another recent study (reference 26) that reported extensive heritable variation in response to the environment. The authors convincingly demonstrate that the conclusions of the previous study were caused by experimental error, so that standing genetic variation was mistaken for acquired (epigenetic) variation. Given the controversy surrounding the possible role of epigenetic variation in mediating phenotypic variation and adaptation, this is an important, clarifying contribution.

[Editors' note: We thank the authors for responding to the reviewers' comments.]

---

## [Author Response]

The following is the authors’ response to the original reviews

**Reviewer #1(Public Review):**
Summary:The authors extended a previous study of selective response to herbivory in Arabidopsis, in order to look specifically for selection on induced epigenetic variation ("Lamarckian evolution"). They found no evidence. In addition, they re-examined result from a previously published study arguing that environmentally induced epigenetic variation was common, and found that these findings were almost certainly artifactual.Strengths:The paper is very clearly written, there is no hype, and the methods used are state-of-the-art.Weaknesses:The result is negative, so the best you can do is put an upper bound on any effects.Significance:Claims about epigenetic inheritance and Lamarckian evolution continue to be made based on very shaky evidence. Convincing negative results are therefore important. In addition, the study presents results that, to this reviewer, suggest that the 2024 paper by Lin et al. [26] should probably be retracted.
**Reviewer #2(Public Review):**
In this paper, the authors examine the extent to which epigenetic variation acquired during a selection treatment (as opposed to standing epigenetic variation) can contribute to adaptation in Arabidopsis. They find weak evidence for such adaptation and few differences in DNA methylation between experimental groups, which contrasts with another recent study (reference 26) that reported extensive heritable variation in response to the environment. The authors convincingly demonstrate that the conclusions of the previous study were caused by experimental error, so that standing genetic variation was mistaken for acquired (epigenetic) variation. Given the controversy surrounding the possible role of epigenetic variation in mediating phenotypic variation and adaptation, this is an important, clarifying contribution.I have a few specific comments about the analysis of DNA methylation:(1) The authors group their methylation analysis by sequence context (CG, CHG, CHH). I feel this is insufficient, because CG methylation can appear in two distinct forms: gene body methylation (gbM), which is CG-only methylation within genes, and transposable element (TE) and TE-like methylation (teM), which typically involves all sequence contexts and generally affects TEs, but can also be found within genes. GbM and teM have distinct epigenetic dynamics, and it is hard to know how methylation patterns are changing during the experiment if gbM and teM are mixed. This can also have downstream consequences (see point below).

We thank Reviewer 2 for this suggestion. We usually separate the three contexts because they are set by different enzymes and not because of the general process or specific function. It would indeed be informative to group DMCs into gbM and teM, but as there are many regions with overlaps between genes and transposons, this also adds some complexity. Given that there were very few DMCs, we wanted to keep it simple. Therefore, we wrote that 87.3% of the DMCs were close to or within genes and that 98.1% were close to and within genes or transposons. Together with the clear overrepresentation of the CG context, this indicates that most of the DMCs were related to gbM. We updated the paragraph and specifically referred to gbM to make this point clearer.

(2) For GO analysis, the authors use all annotated genes as a control. However, most of the methylation differences they observe are likely gbM, and gbM genes are not representative of all genes. The authors' results might therefore be explained purely as a consequence of analyzing gbM genes, and not an enrichment of methylation changes in any particular GO group.

We are grateful to Reviewer #2 for this suggestion. We updated the GO analysis and defined the background as genes with cytosines that we tested for differences in methylation and which also exhibited overall at least 10% methylation (i.e., one cytosine per gene was sufficient). This resulted in a decrease of the background gene set from 34'615 to 18'315 genes. We still detect enrichment of terms related to epigenetic regulation, transport and growth processes. We have updated the corresponding paragraph accordingly.

**Reviewer #1 (Recommendations for The Authors):**
This paper is very clearly written and could be published as-is. The writing could be improved in a few places, for example:"We realized that in this recent study (26), potential errors may have confounded treatments with genetic variation. This is because in that study, Lin and colleagues kept lineages 1-to-1 throughout the experiment by single-seed descent."“This” in the second sentence seems to refer to the confounding, not your realization thereof.I am sure there are more: just give the manuscript a good read-through.

We thank the Reviewer for pointing out that some sentences may not be clear. We have edited the manuscript and focused on avoiding misleading or unclear wording.

**Reviewer #2 (Recommendations for The Authors):**
(1) The authors should distinguish gbM from teM and repeat the GO term analysis with an appropriate set of control genes.

See our response to the public reviews above.

(2) The authors' experimental design should allow them to directly assess whether the rates of epigenetic change are affected by the selective environment. This would require comparison of methylation patterns of individual plants prior to treatment with their progeny (the progeny is what the authors have currently analyzed). This would entail gathering new data, and I don't feel that this analysis is essential, but given the question the authors are addressing (the extent to which a selective environment can induce heritable epigenetic variation), it seems important to test whether the rates of epigenetic change are at all affected by the selection treatment.

While this is a very valuable recommendation, we can currently not address it because the person who gathered the data works at a different university now. However, we keep this in mind for future projects.

Again, we would like to thank the reviewers for the constructive suggestions that help us to improve the manuscript.